# Pim Kinases: Important Regulators of Cardiovascular Disease

**DOI:** 10.3390/ijms241411582

**Published:** 2023-07-18

**Authors:** Sophie Nock, Eima Karim, Amanda J. Unsworth

**Affiliations:** Department of Life Sciences, Faculty of Science and Engineering, Manchester Metropolitan University, Manchester M1 5GD, UK

**Keywords:** Pim kinase, cardiovascular disease, atherosclerosis

## Abstract

Pim Kinases; Pim-1, Pim-2, and Pim-3, are a family of constitutively active serine/threonine kinases, widely associated with cell survival, proliferation, and migration. Historically considered to be functionally redundant, independent roles for the individual isoforms have been described. Whilst most established for their role in cancer progression, there is increasing evidence for wider pathological roles of Pim kinases within the context of cardiovascular disease, including inflammation, thrombosis, and cardiac injury. The Pim kinase isoforms have widespread expression in cardiovascular tissues, including the heart, coronary artery, aorta, and blood, and have been demonstrated to be upregulated in several co-morbidities/risk factors for cardiovascular disease. Pim kinase inhibition may thus be a desirable therapeutic for a multi-targeted approach to treat cardiovascular disease and some of the associated risk factors. In this review, we discuss what is known about Pim kinase expression and activity in cells of the cardiovascular system, identify areas where the role of Pim kinase has yet to be fully explored and characterised and review the suitability of targeting Pim kinase for the prevention and treatment of cardiovascular events in high-risk individuals.

## 1. Introduction

Signal transduction mediated by kinases is crucial in cell communication and in controlling the function and response of cells. Protein phosphorylation, principally on serine, threonine, or tyrosine residues, is one of the most important and widely understood post-translational modifications [1]. Aberrant kinase activity leading to cancer can be caused by mutations; however, in cardiovascular disease (CVD), altered kinase activity may be regulated by an increase in cell stimulation and/or exposure to pathological mediators, as well as protein mutations [2]. With an increased understanding of the contribution of multiple kinases to cardiovascular biology, kinases are becoming more desirable targets for cardiovascular disease. The progression of p38 MAP kinase inhibitors into Phase 3 clinical trials for the prevention of cardiovascular events demonstrates that kinase targeting therapies can be safely administered, but the failure of the inhibitors to improve cardiovascular outcomes [3] identifies that more efficacious kinase targeting therapies are required. Repurposing pre-existing therapies may offer a more advantageous development strategy that quicken the drug development pipeline from synthesis to patient testing.

## 2. Pim Kinases

Proviral integration site for MuLV (Murine Leukaemia Virus) (Pim) Kinases are a family of serine/threonine kinases located in numerous cell types throughout the body [4]. Within the Pim kinase family, there are three isoforms, Pim-1, Pim-2, and Pim-3, and their roles include promoting cell survival [5,6,7,8], proliferation [9], and migration [10]. They are implicated in the pathogenesis of various diseases, with roles identified in multiple cell types, but are most widely known for their contribution to haematological and solid tumour malignancies, which have been reviewed elsewhere [11,12,13,14]. As Pim kinases are constitutively active [15,16], they have become a potential target for cancer therapy, with numerous inhibitors being developed to treat myeloma and myelofibrosis, in addition to other malignancies [17,18,19]. Interestingly, their expression and activity are not limited to cancer, with the Pim kinase isoforms demonstrating widespread expression in cardiovascular tissues (Figure 1). Furthermore, Pim kinases can be upregulated in risk factors associated with CVD. The Pim kinases, therefore, may be alternative therapeutic targets to treat cardiovascular-related diseases and comorbidities.

### 2.1. Structure, Regulation, and Localisation of Pim Kinases

Whilst the three Pim kinase family members are named isoforms due to being highly conserved with high amino acid sequence homology (Figure 2), they are encoded by separate genes and located on different chromosomes (Figure 3). They are classed as serine/threonine kinases, yet the Pim kinases uniquely lack regulatory domains, with only kinase domains present [22]. The kinases do adopt the commonly observed bi-lobed kinase fold structure seen in other kinases; however, the Pim family members contain a novel two-stranded β-sheet predating the αc helix, which distinguishes them from other kinases [23]. Another distinctive feature of Pim kinases is that they contain a unique hinge region [15,23], which makes targeting the kinase family attractive. The hinge region of Pim kinases has two inserted proline residues, widening the ATP (adenosine triphosphate) binding site. This widened ATP site does not allow for the typical hydrogen bonds between ATP and kinases and results in high Km (Michaelis constant) values for ATP [24].

Due to the lack of regulatory domains within the protein (Figure 3) and its protein structure, the Pim kinases are hypothesised to be constitutively active [16,31], and regulation of the kinases is via transcriptional, post-transcriptional, translational, and post-translational mechanisms (Figure 4). Transcriptional regulation of Pim kinases is usually in response to mitogenic stimulants such as platelet-derived growth factor (PDGF) in vascular smooth muscle cells [32,33], high glucose [34], hypoxia, [35,36] and inflammation such as in response to cytokines, IL-6 (interleukin-6) [37], and TNF-α (tumour necrosis factor—alpha) treatment [38]. Furthermore, transcription of Pim-1 has been demonstrated to be regulated downstream of the Janus kinase/signal transducer and activator of the transcription (JAK/STAT) signalling pathway, with activation of the pathway being responsible for its upregulation [39]. Examples of this can be observed downstream of both the erythropoietin (EPO) receptor and thrombopoietin (TPO) receptor (c-Mpl) [40], where EPO and TPO can induce the expression of Pim-1 via JAK/STAT signalling [41,42].

Post-transcriptional modifications are related to mRNA processing. Within the mRNA sequence of each isoform are five AUUA destabilising motifs in the untranslated region (UTR) at 3′ [43,44] (Figure 4), which target the transcripts for rapid degradation [45]. Indeed, this appears to be a key mechanism for the regulation of the Pim kinases, with the half-life of Pim-1 mRNA being approximately 25 min in vascular smooth muscle cells [33].

However, this may not be the sole mechanism of transcriptional regulation of Pim kinase isoform expression levels. In vascular smooth muscle cells (vSMCs), there appear to be other transcriptional methods of control, such as delayed transcription and translation. Cell stimulation with PDGFbb leads to a transient increase in Pim-1 mRNA at 1 h; however, the protein expression profile does not match this time course with protein levels peaking much later at 24 h [33]. In contrast, in response to high glucose, vSMC Pim-1 mRNA and protein levels do correspond to each other, with an increase in both observed at 48 h, suggesting that there may be alternative mechanisms of Pim protein translation depending upon the cellular stimuli [34].

MicroRNAs have also been identified to be important in controlling Pim kinase expression. MicroRNAs interact with the UTR of their target mRNAs, and many have been identified as targets for Pim kinases within the cardiovascular system (summarised in Table 1).

Post-translational modifications of Pim kinase include phosphorylation, dephosphorylation, ubiquitination, and sumoylation. Initially considered to regulate Pim kinase activity, phosphorylation of the Pim-1 isoform on its N terminus [31] was demonstrated instead to control protein stability and degradation rather than catalytic activity. Various candidates have been identified as possible mediators of Pim phosphorylation including Protein Kinase C α (PKCα) and endothelial/epithelial tyrosine kinase (Etk) [53]. PKCα-dependent phosphorylation of Pim-1 has been demonstrated to enhance protein stability, increasing the half-life of the protein [54]. Unphosphorylated Pim-1 is still maintained within the active conformation, with active site residues of the kinase prepared to initiate phosphate transfer [15,31].

Pim kinase autophosphorylation is also believed to be important for kinase stability. Pim-1 was initially thought to be auto phosphorylated on serine 261; however, it has been identified that this occurs on serine 8 [31]. Pim-1 mutants that are unable to autophosphorylate due to mutations within their kinase structure have significantly shorter half-lives, although this could be mediated due to increased clearance [55].

Dephosphorylation and subsequent degradation of Pim kinase occur via protein phosphatase 2A (PP2A), with inhibition of PP2A activity increasing the half-life of Pim kinases [56]. However, it is not clear whether this mechanism is direct or mediated through Suppressors of Cytokine Signalling-1 (SOCS-1). SOCS-1 is involved in the degradation of proteins through ubiquitination [57]. Indeed, SOCS-1 binds to Pim kinases [58], and is required for the PP2A-mediated degradation of Pim kinase [56].

Pim-1 has been shown to be protected from ubiquitin-mediated proteasome degradation following heat shock via a mechanism that involves heat shock protein 90 (HSP90) binding. During the treatment of cells with geldanamycin, an HSP90 inhibitor, rapid degradation and reduced kinase activity of Pim-1 occurs [59,60]. However, ubiquitin-mediated degradation may not extend to all Pim isoforms, as ubiquitination of Pim-2 does not appear to target the kinase for degradation. Adam et al. demonstrate that inhibiting the formation of the cullin RING ligase-containing ubiquitin ligase complex via inhibition of NEDD8-Activating Enzyme did not lead to a modulation of Pim-2 protein levels, despite control proteins being altered. Furthermore, the E1-ubiquitin ligase inhibitor PYR-41 did not alter the stability of the Pim-2 protein [61]. In contrast, a recent study has proposed that ubiquitination of both Pim-1 and Pim-2 is important in regulating the stability of the protein, when under conditions of hypoxia, during hypoxia, deubiquitinase USP28 preferentially binds to Pim-1 and -2 and prevents their degradation [62], suggesting that alternative regulatory mechanisms of Pim kinase degradation may take place under hypoxic conditions.

Degradation of Pim kinases may also occur via Small Ubiquitin-like Modifier (SUMOlyation). Mutation of the potential SUMO site, E171, in Pim-1, increases the protein half-life to almost double that of wild type [55]. Whilst only Pim-1 was studied, Pim-2 and Pim-3 also share similar consensus sequences for the SUMOylation [55]. Furthermore, silencing of the SUMO-targeted E3 ubiquitin ligase RNF4 also increases Pim-1 levels. This suggests ubiquitin-facilitated proteasomal regulation of Pim kinase levels [55].

### 2.2. Pim-1

Of the three Pim kinase isoforms, Pim-1 is the most extensively researched family member. The Pim-1 gene is located on chromosome 6, and in-frame initiation translation codons on the chromosome result in two Pim-1 isoforms of different sizes [63]. Pim-1L (44 kDa) is an N-terminal extension of Pim-1S (34 kDa), initiated by an upstream CUG codon [44,64], as shown in Figure 3. Both isoforms have comparable kinase activity [44], but due to the extra proline-rich motif in Pim-1L, it is suggested that Pim-1L primarily localises at the plasma membrane, with Pim-1S being primarily nuclear or cytosolic [65]. There is little literature focusing on the differing isoforms of Pim-1; however, one study has identified that there may be subtle differences between their phosphorylation activities within the context of cancer [66].

Pim-1L and Pim-1S can phosphorylate the androgen receptor at serine-213, whereas only the larger isoform could phosphorylate at threonine 850 in vivo suggesting some differential phosphorylation, which may be related to their localisation within the cell [66].

Pim-1 expression is widely associated with hematopoietic neoplasia and solid tumours [11,22], and increased Pim-1 expression is typically a marker of poor prognosis [67]. Pim-1 inactivates pro-apoptotic protein BAD by phosphorylating it on Ser112 [5]. Furthermore, it promotes cell cycle progression through phosphorylation and activation of Cdc25 [68,69], and phosphorylation and inhibition of Cdc25-associated kinase 1 (C-TAK1) [70], an inhibitor of Cdc25C. The promotion of tumorigenesis through phosphorylation of c-Myc [71,72] and regulation of mTORC [73] is also be mediated via Pim-1. Pim-1 has also been shown to promote cell migration and chemotaxis (metastasis) via the phosphorylation of CXCR4 [74].

### 2.3. Pim-2

Pim-2, located on the X chromosome, is the most divergent member of the kinase family (Figure 2). Similar to Pim-1, Pim-2 also has multiple isoforms, 34, 37, and 40 kDa (Figure 3) [75]; however, it only shares 66% amino acid sequence homology with Pim-1, with the most substantial differences being within the C terminal (23 residues) (Figure 2). Within this sequence are six proline residues that are not predicted to form a helical structure; therefore, Pim-2 lacks a C-terminal α helix [24] compared to the other isoforms. This lack of helical structure is thought to increase the flexibility of the terminus, resulting in structural changes to Pim-2 compared with Pim-1 and -3, which may offer an explanation as to why pan-Pim kinase inhibitors appear to be less potent on Pim-2 compared to the other Pim family members [76,77]. Modelling approaches have demonstrated that amino acids D128 and E171 are required for potent inhibition of Pim-2 but not Pim-1 and -3 [29]. Despite these apparent differences in structure, Pim-2, like Pim-1, can also phosphorylate c-Myc [72] and maintain mTORC signalling by maintaining inhibitory phosphorylation on eukaryotic initiation factor 4E binding protein (4E-BP1) and BAD, allowing for cell growth and survival [16].

All three Pim-2 isoforms are cytoplasmic; however, they do have different levels of expression and protein stability. 37 kDa and 34 kDa isoforms of Pim-2 have very short half-lives (less than 30 min), whereas the 40 kDa isoform has a half-life approximately twice that of the shorter isoforms [61]. These differences in protein stability are likely due to the disordered N-terminal domains that can be recognised directly by the proteasome [61].

Differences in the activity of the Pim-2 isoforms have also been observed in vitro. The largest isoform has been shown to be less active when compared to the smaller isoforms; however, it is unclear why this is; one hypothesis is that the larger 40 kDa isoform may contain an auto-inhibitory sequence [7].

### 2.4. Pim-3

Pim-3 is located on chromosome 22 and shares 77% sequence homology and similar protein structure with Pim-1. However, unlike the other family members, which are expressed as multiple-length variants, Pim-3 is only expressed as one 34 kDa isoform (Figure 3) [78]. Similar to Pim-1 and Pim-2, Pim-3 can phosphorylate BAD at Ser112, with Pim-3 knockdown shown to reduce phosphorylation at this site [8,79]. MYC activity can also be augmented by Pim-3, along with control of protein synthesis [80].

The summary of the key differences between the different members of the Pim kinase family can be found in Table 2.

## 3. Pim Kinase and Cardiovascular Disease

Whilst most well known for their role in cancer progression, there is increasing evidence for wider pathological roles of Pim kinases, including in CVD, a leading cause of mortality worldwide [88]. Atherosclerotic plaque rupture or erosion activates circulating platelets, resulting in activation of the coagulation cascade and thrombosis. Plaque dynamics are regulated by multiple cell types, including endothelial cells, vascular smooth muscle cells (VSMCs), monocytes, and platelets. The formation of an occlusive thrombus that blocks an artery can subsequently lead to myocardial infarction (MI).

The Pim kinase isoforms have a widespread expression in cardiovascular tissues, including the heart, coronary artery, aorta, and blood (Figure 1), and have been demonstrated to be upregulated in a number of co-morbidities/risk factors for cardiovascular disease such as smoking and hyperglycaemia [34,89]. Pim-1 is increased 11-fold in coronary artery disease [90], present in pulmonary arterial hypertension [91], and Pim-2 is upregulated in atherosclerotic arteries in coronary artery disease [92]. This suggests that Pim kinase inhibition may be a desirable therapeutic for a multi-targeted approach to prevent and treat cardiovascular disease.

### 3.1. Hypertension

Pulmonary arterial hypertension (PAH) is defined as high blood pressure within the lungs. In PAH, Pim-1 has been identified as a potential biomarker, with plasma Pim-1 levels correlating directly to disease severity [91], and increased expression of Pim-1 was observed in PAH patients’ pulmonary artery smooth muscle cells (PASMCs), compared to healthy controls [93].

#### 3.1.1. Vascular Smooth Muscle Cells

Vascular smooth muscle cells (vSMCs) are found in the arterial walls and are crucial to maintaining arterial wall integrity and regulating vascular tone. Cellular DNA damage and its repair in PASMCs are one of the main contributors to the causes of PAH pathology [94]. Usually, cells undergoing DNA damage are destined for apoptosis if they are inefficiently repaired [94]. In cancer cells, Pim kinases have been demonstrated to regulate non-homologous end joining, allowing for cells to pass DNA damage checkpoints, evading apoptosis [95], and this can also be observed in PASMCs. In vitro Pim-1 inhibition in PASMCs using SG1-1776 decreases DNA damage repair, proliferation and induces apoptosis [93]. In vivo mice experiments demonstrated, using two different Pim kinase pharmacological inhibitors and two different models of PAH, that Pim kinase inhibition significantly reduces pulmonary pressure, demonstrating the potential of Pim kinase inhibition to be used as a therapeutic for PAH [93].

In addition to PAH, exposure to high glucose in culture, similar to those experienced in hyperglycaemia and diabetes, has also been shown to induce Pim-1 mRNA and protein expression in VSMCs. Wang et al., using both in vitro and in vivo models, increased Pim-1 expression in the tunica media of hyperglycaemic rats [34]. The same group also previously demonstrated that high glucose induces VSMC proliferation, a feature of PAH, via an increase in CXCR4 signalling [96]. Whilst the authors demonstrated that this was due to the increased expression of SDF-1α, Pim-1 has been shown to positively regulate CXCR4 signalling in other cell types [97], indicating the potential to target the kinase to prevent CXCR4-mediated VSMC proliferation during hyperglycaemia.

Roles for both Pim-1 and Pim-3 have also been proposed in the regulation of VSMC contractility via phosphorylation of the myosin targeting subunit (MYPT1) [98]. MYPT1 phosphorylation inactivates myosin light-chain phosphatase, suppressing myosin dephosphorylation and causing hypercontractility [99]. Hypercontractility of VSMCs contributes to vascular remodelling, leading to rigid and stiff vessels, contributing to hypertension. This suggests that the inhibition of Pim kinases may be beneficial to reduce hypertension.

#### 3.1.2. Endothelial Cells

Endothelial cells play key roles in the regulation of VSMCs and hypertension, particularly via the production of nitric oxide, which causes vasodilation of VSMCs [100]. Studies using human umbilical vein endothelial cells (HUVECs) have identified a role for Pim in the regulation of VEGF-dependent endothelial nitric oxide synthase (eNOS) signalling downstream of the VEGF receptor Flk1 [101]. siRNA-mediated knockdown of Pim-1 in HUVECs reduces VEGF-induced eNOS phosphorylation at Ser-663 [101]. In addition, treatment of HUVECs with Pim kinase inhibitor SMI-4a reduces VEGF-mediated nitric oxide (NO) production after 24 h of treatment. Inhibition of Pim kinases under these conditions may result in a reduction of nitric oxide-mediated relaxation of the vessel and promote hypertension.

Further research is thus required to investigate the impact of Pim kinase inhibition on the underlying mechanisms and relationship between VSMCs and ECs that control vasodilation and vasoconstriction.

### 3.2. Atherosclerosis

The formation of atherosclerotic plaques can partially be attributed to altered cholesterol levels within the blood, increasing the amount of foam cells and fatty deposits within the arterial wall.

#### 3.2.1. Cholesterol Metabolism

Studies using Pim-1 deficient mice have identified a negative regulatory role for Pim-1 in atherosclerotic plaque formation. Pim-1 has been identified to phosphorylate and regulate ATP-binding cassette transporter A1 (ABCA1) and subsequently enhance reverse cholesterol transport, preventing the formation of foam cells [102]. siRNA-mediated deletion of Pim-1 in ABCA1-expressing hepatocytes was found to associate with decreased levels of ABCA1 present on the surface of HEP2G cells [102], resulting in lower levels of cellular high-density lipoprotein (HDL) and HDL efflux and an increased risk of cardiovascular events [103,104]. In support of this, Pim-1 KO mice were found to have less plasma HDL [102], and an earlier study investigating the effects of increasing levels of mIR-33, which targets and decreases Pim-1 levels, [105] demonstrated decreased plasma levels of HDL in mIR-33-treated mice [106].

#### 3.2.2. Endothelial Cells and Atherosclerosis

Endothelial cell dysfunction is a contributor to the formation of atherosclerosis. Increased vascular permeability supports monocyte transmigration and differentiation into macrophages and foam cells, leading to the promotion of atherosclerotic plaque development [107]. Inflammatory cytokines derived from the plaque and other vascular cells increase vascular permeability via the formation of actin stress fibres, filopodia, and membrane ruffles in endothelial cells [38]. Cytokines can also influence endothelial cell migration [108] and angiogenesis [109]. Endothelial cell angiogenesis-driven neovascularisation of plaque material contributes to atherosclerotic plaque growth and instability [110,111]. Indeed, neovascularisation of plaque regions is associated with a 1.5-fold increased risk of cardiovascular events compared to plaque regions with lower levels of vessel density [112].

Although the endogenous expression of Pim-1 in HUVECs appears low, it has been shown to increase in response to VEGF treatment [101] and following pharmacological inhibition of PI3K [113].

Quiescent endothelial cells rapidly switch to an activated state gaining the ability to sprout, migrate, and proliferate [114] in response to VEGF released from VMSCs and macrophages [115]. Pim-1 has been shown to mediate VEGF responses in mice endothelial precursor cells [116]. Silencing of Pim-1 using a lentiviral vector expressing Pim-1 siRNA causes a significant reduction in PECAM-1 after 5.5 days after differentiation, with VE-cadherin surface levels also being reduced at a later time point of 8.5 days [116]. Pim-3 has also been shown to mediate inflammatory TNF-α signalling in endothelial cells, with TNF-α treatment associated with increased Pim-3 mRNA expression in HUVECs [38] in vitro and Pim-3 silencing attenuating endothelial cell sprouting in vivo [38]. These findings support a positive role for Pim-3 in TNF-α inflammation-mediated cytoskeletal rearrangements, vascular permeability (cell detachment), and angiogenesis, thereby demonstrating the potential of targeting Pim-3 to protect against pathological endothelial cell activation to reduce vascular permeability and neovascularisation.

#### 3.2.3. Monocytes and Macrophages

Transmigration of monocytes and their differentiation into macrophages is a key step in atherosclerotic plaque development, promoting the formation of unstable plaques and enhancing the proinflammatory environment. In macrophages, Pim-2 mediates suppression of the mTORC1 pathway restraining inflammation [117]. Macrophages overexpress Pim-2 in response to stimulation by oxidised low-density lipoprotein (LDL) and enhanced Pim-2 expression is observed in macrophages from Apo-E-deficient mice compared to C57BL/6 controls [117]. In support of these findings, overexpression and knockdown of Pim-2 in mouse models demonstrated a decrease and increase in aortic plaque regions, respectively, suggesting that Pim-2, in contrast to that observed for Pim-1 that lowers plasma cholesterol [102], through suppression of the mTORC1 pathway, is anti-atherogenic and protective in hyperlipidaemic conditions [117].

#### 3.2.4. Vascular Smooth Muscle Cells and Atherosclerosis

Aberrant VSMC proliferation also contributes to initial plaque formation by increasing the mass and width of the atherosclerotic plaque. They may also contribute to vessel hardening, inflammation, and development of the necrotic atherosclerotic core, by secretion of extracellular matrix and cytokines, respectively [118]. Pharmacological inhibition of Pim kinases results in decreased phosphorylation of BAD, leading to the initiation of VSMC apoptosis [34], mirroring observations made in cancer cells [5,8]. Similarly, Paulin et al. observed decreased phosphorylation of BAD (Ser112) and enhanced apoptosis in PAH PASMCs treated with siRNA targeting Pim-1 [119]. VSMC proliferation was also found to be reduced following treatment with Pim-1 shRNA [46], identifying a potential target for the inhibition of initial plaque formation. However, it has been proposed that VSMC proliferation may be beneficial for atherosclerotic plaque stability in advanced lesions [118]; therefore, inhibition of Pim kinases may not be a desirable therapeutic at advanced stages of plaque development, if it leads to VSMC cell death.

### 3.3. Thrombosis

During atherosclerotic plaque rupture or erosion, endothelial cells detach from the vascular wall and contribute to the pathogenesis of atherothrombosis, where exposure of atherosclerotic plaque components and the underlying extracellular matrix proteins, and endothelial cell damage precipitate platelet activation and thrombus formation.

#### 3.3.1. Endothelial Cells and Thrombosis

Walpen et al. demonstrated that Pim-1 regulates the adhesive phenotype of mouse aortic endothelial cells, with the loss of Pim-1 associated with increased endothelial cell adhesion [120]. In addition, there also appears to be a role for Pim-3 in endothelial cell attachment. RNAi silencing of Pim-3 reduces HUVEC spreading and migration [121] but does not alter endothelial adhesion to cell matrices In vitro. The mechanisms behind the contribution of Pim-1 and Pim-3 to increased endothelial cell attachment and spreading may be related to their function in relation to the cytoskeleton. Pim-1 has been shown to regulate cytoskeletal organisation in cancer cells [10,122], siRNA mediated Pim-1 knockdown in HUVECs has been shown to attenuate endothelial cell sprouting [116], but more work is required to elucidate the specific role of Pim-1 in this regard. Pim-3 has been observed to be localised at the leading edge of lamellipodia in specialised actin focal complexes within endothelial cells, supporting its role in the regulation of the actin cytoskeleton. Pim-3 silencing was subsequently shown to reduce the ability of endothelial cells to form stress fibres, Ref. [121] suggesting that Pim-3 plays a role in regulating the endothelial cytoskeleton. The inhibition of Pim kinases in this context may be beneficial as it may result in reduced detachment of endothelial cells, limiting the exposure of thrombogenic material to the blood.

#### 3.3.2. Platelets

Pim kinases have also been shown to play a role in mediating circulating platelet functions. Platelets are key initiators of arterial thrombosis following exposure to the sub-endothelium and atherosclerotic plaque contents, whereby platelets adhere and aggregate forming an occlusive thrombus [123,124], which can lead to myocardial infarction and cardiac death. Our study investigating the role of Pim kinase in platelet function identified anti-platelet properties of pan-Pim kinase inhibitors. Treatment of human platelets with AZD1208 caused a reduction in surface expression levels of the thromboxane A2 receptor (TPaR), leading to a reduction in signalling events downstream of TP-coupled G proteins (Gq and Ga13), leading to reduced platelet activation [125].

In addition to regulation of TPaR-dependent platelet responses, Pim kinase inhibition also attenuates CXCR4-mediated platelet responses, with platelet aggregation to CXCR4; ligand SDF-1a also reduced following treatment with AZD1208. These observations are supported by other studies demonstrating that Pim-1 mediates regulation of CXCR4 signalling in CLL and Jurkat cells [97]. These findings indicate that Pim-1 inhibition could be a novel mechanism for anti-platelet target therapies. This is supported by observations from thrombosis studies using Pim-1 deficient mice, which show reduced thrombus formation using in vitro models and reduced thrombosis in vivo following acute treatment of mice with a pan-Pim kinase inhibitor AZD1208 [125]. The roles of Pim-2 and Pim-3 in platelet function have yet to be described. Despite Pim-1 mice demonstrating reduced thrombotic capacity, normal haemostatic functions were maintained, suggesting that Pim-1 inhibition may be a suitable target for antiplatelet therapy, having no adverse effects on bleeding [125].

In addition to platelet function, higher platelet counts and thrombocytosis are strongly associated with cardiovascular disease [126]. Studies using mice demonstrate that individual deletion of either Pim-1 or Pim-2 does not alter platelet number [127,128], whereas triple deletion of all three Pim kinases in mice has been associated with reduced platelet count [43,129], suggesting some but not total compensation between the isoforms. In support of roles for the Pim kinases in haematopoiesis, bone marrow hematopoietic stem cells in triple Pim KO mice show a reduced ability to differentiate into haematopoietic cell lineages including megakaryocytes in vitro [43] and have reduced gene expression of myeloid lineage transcription factor GATA1 [128]. This is supported by observations that Pim-1 mRNA is upregulated in response to thrombopoietin, [128] and erythropoietin [42], mediators of thrombopoiesis and erythropoiesis, respectively. Observations of thrombocytopenia in clinical trials using pan-Pim kinase inhibitors to treat varying malignancies [17,18,19] also further provide evidence that Pim kinases mediate platelet production; however, reports vary regarding the extent of reduced platelet count, with one study reporting no thrombocytopenia events [130] or occurring in a small subset of patients [17] (Table 3).

Cortes et al. describe no reports of thrombocytopenia following treatment of acute myeloid leukaemia patients with AZD1208 (ClinicalTrials.gov, NCT01489722) but did observe thrombocytopenia in 14.3% of solid tumour patients [17]. Furthermore, two Phase I multiple myeloma trials for PIM447 (ClinicalTrials.gov; NCT01456689, NCT02160951) observed 32.9% [19] and 53.8% [18] incidence rates of grade 3/4 levels of thrombocytopenia, with additional patients experiencing reduced platelet counts [18,19]. In both trials, a reduction in therapeutic dose or cessation of treatment facilitated the recovery of platelet count. These observations demonstrate that patient response to drug therapy may be affected by disease-specific pathologies with thrombocytopenia as a common complication in patients with solid tumours [131] and multiple myeloma [132].

Interestingly, a recent study investigating the potential of using Pim kinase inhibitors for the treatment of myelofibrosis demonstrated that pan-Pim kinase inhibitor TP3654 (ClinicalTrials.gov; NCT04176198, accessed on 21 April 2023) reduced myelofibrosis mediated thrombocytosis (increased platelet count) in mouse models [133], highlighting the potential suitability of Pim kinase inhibitors for the treatment of conditions associated with aberrant platelet production, turnover, and thrombocytosis. Further studies are needed to elucidate the role of Pim kinase in platelet production and determine the cardiotoxicity profiles of Pim kinase inhibitors in disease-relevant cohorts.

**Table 3 ijms-24-11582-t003:** Cardiovascular side effects related to Pim kinase inhibitor therapy.

Drug	Disease	Number of Patients	Cardiovascular Side Effects (% of Patients)	Clinical Trial Number/Reference
AZD1208	Solid Tumour	67	Thrombocytopenia (5%) Platelet count decreased (25.7%) 3% atrial fibrillation	NCT01588548 [17]
AZD1208	Acute myeloid leukaemia	55	Hypotension (31.3%)	NCT01489722 [17]
Pim447	Multiple myeloma	13	Thrombocytopenia (76.9%) Electrocardiogram QT prolonged (15.4%)	NCT02160951 [18]
LGH447 (Pim447)	Multiple Myeloma	77	Thrombocytopenia (32.9%) Congestive cardiomyopathy (2.7%) Palpitations (1%) Atrial fibrillation (2.5%) Bradycardia (1%) Sinus bradycardia (2.5%) Tachycardia (53%)	NCT01456689 [19]
LGH447 (Pim447)	AML or High risk Myelodysplastic syndrome	70	Thrombocytopenia (32.4%) Platelet count decreased (9%) Disseminated intravascular coagulation (18%) Tachycardia (9%) Sinus bradycardia (9%) Electrocardiogram Qt Prolonged (13.5%) Hypertension (9%) Hypotension (18.8%)	NCT02078609 [134]
SGI-1776	Relapsed/ Refractory Leukaemia	14	Prolonged QtC time Studies withdrawn with no results	NCT01239108 NCT00848601 [135]
TP3654	Advanced solid tumours	22	No reported cardiovascular side effects	NCT03715504 [130]
TP3654	Myelofibrosis	8	No reported cardiovascular side effects	NCT04176198 [136]
INCB053914	Advanced haematological malignancies	8	Thrombocytopenia (15%)	NCT02587598 [137]

### 3.4. Cardiac Dysfunction, Injury, and Recovery

Cardiomyocyte dysfunction, interstitial fibrosis, and an increase in cardiac size without an increase in cellular number can lead to hypertrophy, decreased pump function, and ultimately heart failure [138]. One of the main causes of heart failure may be also related to myocardial cellular death post-myocardial infarction occurring from oxygen cessation through coronary artery occlusion [139]. Myocardial infarcts primarily occur after the occlusion of the coronary artery [140], principally through narrowing via atherosclerosis, plaque rupture, or erosion, leading to platelet activation and thrombosis [141,142].

#### Cardiomyocytes

Pim-1 is expressed in the neonatal heart, but its expression diminishes during postnatal development [143,144]. In neonatal mice, Pim-1 is primarily localised within the nucleus of cardiomyocytes, where, via regulation of NFATc1, it mediates cardiac development and relocates to the cytosol by 30 weeks [143]. In contrast to Pim-1, the levels of Pim-2 expression in mouse cardiomyocytes do not change as the mouse ages, whereas Pim-3 has the opposite effect and increases with age [143].

Pim-1 has been demonstrated to play a wide range of roles within cardiomyocytes. Overexpression of Pim-1 in c-kit positive cardiac cells promotes proliferation and survival, whilst decreasing senescence [145]. Furthermore, overexpression of Pim-1 within transgenic hearts results in significantly increased calcium dynamics, due to increased expression of sarcoendoplasmic reticulum Ca2+ATP-ase (SERCA) [143]. Pim-1 also has roles in the protection of the mitochondria within the heart, such as prevention of mitochondrial fission [144,146] via the regulation of DRP1. Taken together, the protection of the mitochondria, regulation of calcium transient measurements, and the promotion of proliferation via Pim kinases have the potential to ‘rejuvenate’ cardiac stem cells and may prevent damage after ischemic injury [145].

A role for Pim-1 has also been demonstrated in telomere lengthening. Overexpression of Pim-1 in murine cardiomyocytes resulted in an increase in telomere length compared to controls [147]. Decreased telomere length is a hallmark of heart failure due to cells entering senescence [148]; therefore, Pim-1 plays a protective role in cardiomyocytes.

Pim-1 has been identified in the upregulation of c-Kit in haematopoietic stem cells [149]. This can also be observed in cardiomyocytes, where overexpression of Pim-1 increases the level of c-Kit [150]. C-Kit already plays a role in cardioprotection [151], but it has been demonstrated that Pim-1 has protective roles during hypoxia independently of c-Kit activity [150], promoting survival.

In addition to its contribution to cardiac metabolism, Pim-1 is also responsible for phosphorylating cardiac troponin I (cTnI) [152]. Reduced cTnI phosphorylation results in lack of heart contractility and is a key contributor to diabetic cardiomyopathy. Indeed, Pim-1 has also been shown to be downregulated in the cardiac tissue in diabetic mice [101]. Interestingly, cardiac-specific overexpression of Pim-1 and increased phosphorylated CTnI have been shown to decrease infarct size and maintain heart contractility after myocardial infarction [152].

It has also been demonstrated that Pim-1 expression in cardiomyocytes is upregulated in vivo, with increased Pim-1 expression observed in heart tissue sampled from patients with heart failure [143]. Mohsin et al. propose this to be an attempt to rejuvenate and protect the heart from failure [145], as Pim-1 is induced in response to cardioprotective agents such as insulin-like growth factor (IGF-1), dexamethasone, and phorbol 12-myristate 13-acetate (PMA) in vitro, demonstrating protective and reparative properties of the kinase [143]. Furthermore, transgenic expression of Pim-1 in the myocardium protected mice from infarction injury, and inhibited cardiomyocyte apoptosis and hypertrophy [143,153].

Interestingly, in contrast to findings identifying Pim-1 as a key mediator of cardiomyocyte function and survival, genetic deletion of Pim-1 in mice does not appear to lead to cardiomyopathy in basal conditions. The haemodynamics of Pim-1 deficient mice hearts are comparable to wild-type controls, despite an increase in apoptotic myocytes [143]. This suggests potential redundancy between the isoforms, with Pim-2 and -3 compensating for the loss of Pim-1 in basal conditions to prevent cardiomyopathy, with loss of cardiac Pim-1 associated with increased expression of Pim-2. This indicates that under healthy conditions, either Pim-2 or Pim-3 activity can compensate for Pim-1, or that Pim-1 does not play a role in cardiomyocyte function. Interestingly, loss of Pim-1 is considered to be detrimental when murine hearts are challenged through injuries such as MI or transaortic constriction (TAC), with Pim-1-deficient mice displaying decreased contractility and increased apoptotic death, despite a corresponding upregulation in Pim-3 [143,153]. This suggests that under conditions of cardiac stress, Pim-1 is cardioprotective, but Pim-2 and -3 are not and are unable to compensate for the loss of Pim-1. Further evidence for this is Pim triple knockout mice experiencing premature cardiac ageing and progression towards heart failure at 6 months [144], highlighting the drawbacks of complete deletion or total inhibition of Pim kinase function. These observations will also have ramifications for drugs affecting upstream regulators of Pim kinase.

These in vivo findings in mice are further supported by reports made in the first Pim kinase inhibitor Phase I clinical trial for SGI-1776 (NCT01239108) (a Pim kinase and FLT3-D3 inhibitor), which was terminated early and withdrawn due to QTc prolongation [135], which may be related to linked abnormal calcium handling via Pim regulation of SERCA [143]. However, it should be noted that SGI-1776 is described as an inhibitor of both FLT3 and Pim kinase. Phase I trials for other Pim kinase inhibitors have yielded conflicting results (Table 3). Trials for AZD1208 do not report prolonged QTc time [17], whilst results regarding myocardial side effects are conflicting with PIM447 [18,19]. These differences in observations could be a side effect of disease severity and patient demographics and should be investigated further due to different levels of Pim kinase inhibition. These observations in preclinical studies and clinical trials indicate that Pim kinase expression and function are important for cardioprotection. Further dosing studies are required to determine whether a reduction in Pim kinase activity, rather than total ablation of kinase expression and activity, can be tolerated by patients and whether this inhibition can be balanced with the role of the Pim kinases in cardiac repair post-ischaemia.

## 4. Conclusions

Within the cardiovascular system, the Pim kinase family members play divergent roles, mediating various cell types that are summarised in Figure 5. Expression of Pim kinases is upregulated in response to cardiac and cardiovascular damage, mediating proliferation, inflammatory responses, migration, and survival of the cells under pathological conditions, providing protective effects. Whilst proliferative and survival signals are required for cell repair and basal protection, particularly in cardiomyocytes, deregulated growth mediated by Pim kinase and activation of VSMCs, endothelial cells, and platelets contribute to atherosclerosis and hypertension through narrowing of the blood vessels, macrophage activation, increased plaque formation, endothelial activation, and thrombosis.

Drug therapy targeting Pim kinases and reducing Pim kinase isoform activity has the potential to offer a multifaceted, multi-cellular approach to combat cardiovascular disease. However, current clinical trials have raised some concerns regarding the cardio-safety of current Pim kinase therapies (Table 3) that require further investigation. Future development must focus on fully elucidating the individual roles of the three Pim kinase isoforms within the cardiovascular system, alongside developing isoform-specific targeting strategies to ensure that the positive anti-atherosclerotic, anti-hypertensive, and anti-thrombotic properties of Pim kinase inhibitors can be appropriately balanced to prevent the cardiac side effects observed with total Pim kinase removal and observed with some existing Pim kinase inhibitors.

## Figures and Tables

**Figure 1 ijms-24-11582-f001:**
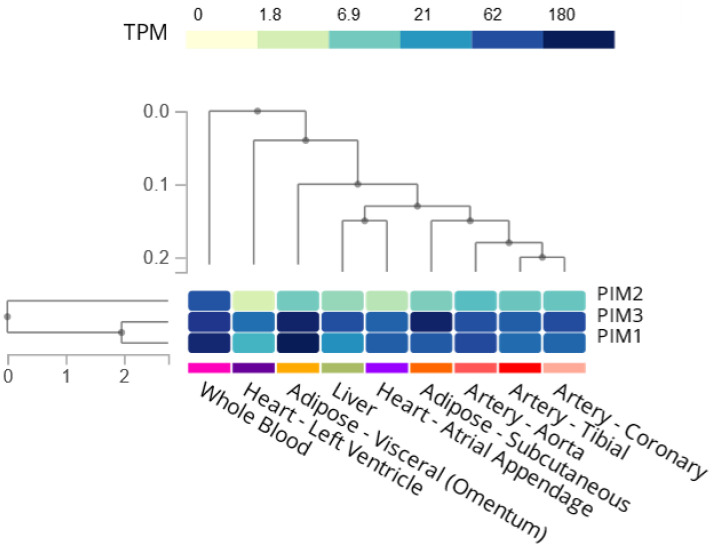
Bulk tissue gene expression of Pim kinases in tissues relating to the cardiovascular system. Created using the GTEX portal [20,21]. Darker colour refers to more transcripts per million (TPM).

**Figure 2 ijms-24-11582-f002:**
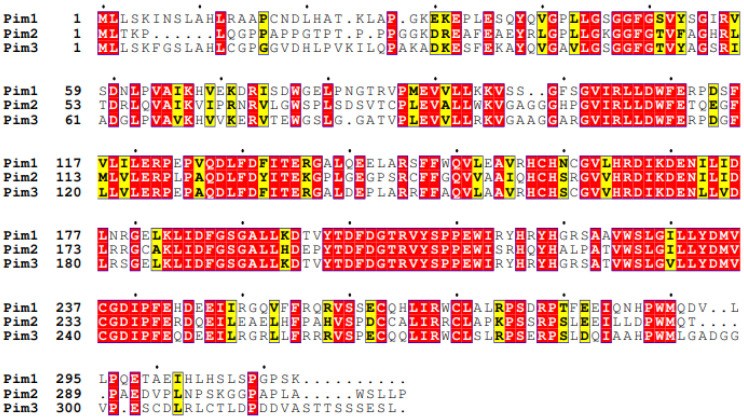
Sequence alignment of Human Pim family kinases. Sequences were aligned using Clustal Omega [25] and ESPript 3 [26] using Uniprot accession sequences Pim-1—P11309, Pim-2—Q9P1W9, and Pim-3—Q86V86. Exact matches are shown in white text on a red background, and similar residues in boxed yellow and black text on a white background are little consensus between the three isoforms.

**Figure 3 ijms-24-11582-f003:**
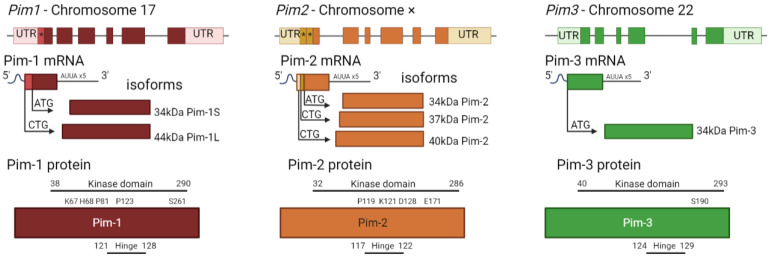
Pim kinase family gene structures and corresponding protein structures. Each gene has six codons surrounded by large untranslated regions (UTR). Alternative start codons within the gene (*) correspond to alternate mRNA transcripts, representing the different isoforms. Each isoform consists of only a serine/threonine kinase domain, with molecular weights varying from 34–44 kDa. The largest isoform is shown in more detail as to where the kinase domain and hinge regions are present and where key sites are located (not to scale). Pim-1 key sites include K67, H68 (site mutation increases kinase activity [23]), P81 (site mutation decreases kinase activity [23]), P123 (proline region in hinge site which is unique to Pim family kinases [11,15]), and S261 (phosphorylation site [27]). Pim-2 key sites include P119 (proline residue in hinge site, unique to Pim kinases (Q9P1W9—[28]), K121 (site mutation renders kinase inactive [16]), D128 and E171 (required for potent drug inhibition [29]). Pim 3 sites are the hinge region (Q86V86 [28]) and S190 (predicted autophosphorylation site [30]).

**Figure 4 ijms-24-11582-f004:**
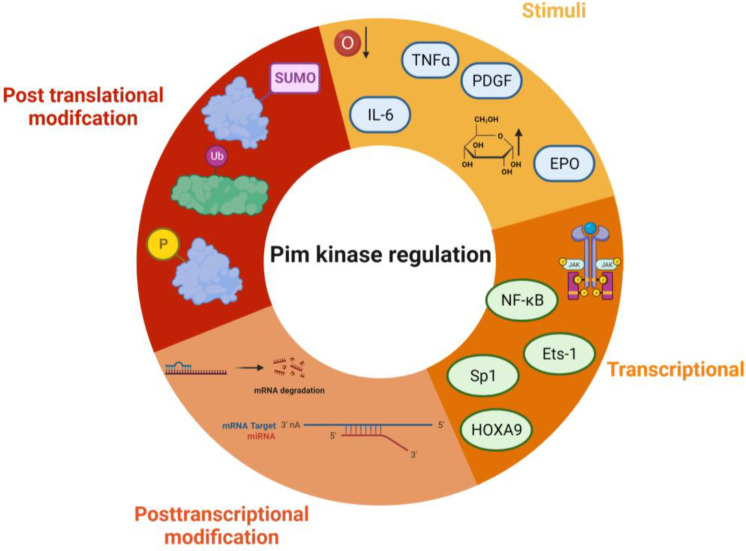
Regulation of Pim kinases. Some of the key regulatory pathways that control Pim kinase activity and signalling. Stimuli—TNFα—Tumour necrosis factor alpha, EPO—erythropoietin, IL-6—interleukin 6, PDGF—platelet-derived growth factor, hypoxia, and high glucose. Transcriptional modifications include NF-Kb—Nuclear factor kappa B, Ets-1—ETS proto-oncogene 1, Sp1—Specificity Protein 1, and HOXA9—Homeobox A9. Post-translational modifications include ubiquitination, sumoylation, and phosphorylation. Post-transcriptional modification—mRNA degradation, and miRNA binding. The figure was created using Biorender.

**Figure 5 ijms-24-11582-f005:**
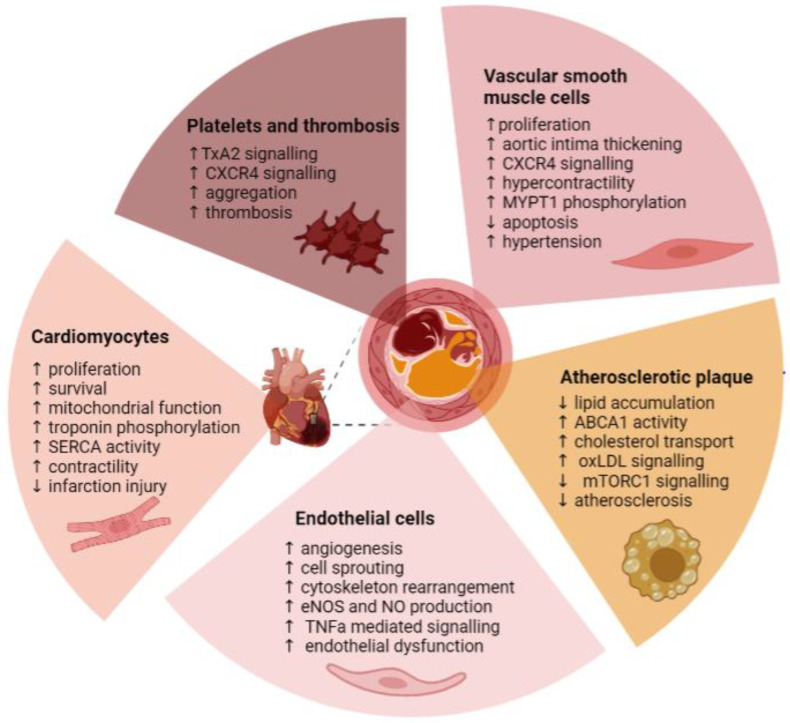
Summary of the role of Pim kinases within the cardiovascular system. TxA2—thromboxane A2; MYPT1—myosin phosphatase target subunit 1; ABCA1—ATP-binding cassette subfamily A member 1; oxLDL—oxidised low-density lipoprotein; mTORC1—mammalian target of rapamycin complex 1; eNOS—endothelial nitric oxide synthase; NO—nitric oxide; TNF-α—tumour necrosis factor alpha; SERCA—sarco(endo)plasmic reticulum calcium transport ATPase; ↑–increase; ↓–decrease. Figure created using Biorender.

**Table 1 ijms-24-11582-t001:** MicroRNAs targeting Pim kinase in cardiovascular cell types.

MicroRNA	Target	Cell Type	Result	Ref
MiR-328	Pim-1	Pulmonary artery smooth muscle cells. Human umbilical vein endothelial cells	Overexpression of MiR-328 increases the level of Pim-1 and promotes proliferation. Inhibition of MiRNA-328 promotes angiogenesis in high glucose conditions by increasing Pim-1 expression.	[46,47]
MiR-214	Pim-1	Bone marrow-derived mesenchymal stem cells Vascular smooth muscle cells	PDGF increases the expression of this MiRNA, resulting in Pim-1 knockdown.	[32]
MiR-206	Pim-1	Mesenchymal stem cells in hypoxia Pulmonary artery smooth muscle cells.	Reduction in miR-206 causes increased proliferation and reduced apoptosis, which is protective in infarcted hearts.	[36,48]
Mir-1	Pim-1	Vascular smooth muscle cells	Myocardin increases the expression of Mir-1, which results in the inhibition of VSMC proliferation via the knockdown of Pim-1.	[49]
miR 410-5-p	Pim-1	Diabetic cardiomyopathy	MiR-410-5p inhibition decreases high glucose-induced myocardial apoptosis, preventing cardiomyopathy through the upregulation of Pim-1.	[50]
MiR-195	Pim-1	Sepsis—endothelial cells	Inhibition of miR-195 results in increased Pim-1, which is protective in a model of sepsis.	[51,52]

**Table 2 ijms-24-11582-t002:** Comparison of the Pim kinase family members.

	Pim-1	Pim-2	Pim-3
Isoforms	44 kDa 34 kDa [42,64]	34 kDa 37 kDa 40 kDa [7,61,73]	34 kDa [78]
Isoform differences	Preferential of targets as opposed to others for Pim-1L and Pim-1S [66] such as Pim-1S favouring the androgen receptor [81] and Pim-1L phosphorylating Etk [65]	Largest isoform less active [7,61]	N/A
Protein localisation	Pim-1L plasma membrane Pim-1S cytosolic [65]	Cytoplasmic [61]	Cytoplasm [82]
Chromosome [83]	6 [63]	X [83]	22 [83]
Sequence similarity [24,78]	Pim-1 Pim-2 66%, Pim-1 Pim-3 77% [82], Pim-2 Pim-3 44% (Figure 2)		
Structural differences		C-terminal helix missing in Pim-2 Six proline residues in the last structure [24].	
Transcriptional regulation differences	NF-KappaB [84]. HOX9A upregulates Pim-1 [85]		Not dependent on NF-KappaB, but Sp1 and Ets-1 [86]. HOX9A downregulates Pim-3 [87].
Degradation	Ubiquitination [59,60,62]	Ubiquitin independent in normoxia [61]. Under hypoxia, ubiquitination takes place [62]	Believed to be Ubiquitinated similarly to Pim-1 [62]
Consensus peptide sequence (AKRRRRHPSGPPTA) binding affinity [31]	40–60 nM	640 nmol/L	40–60 nM
Kd for consensus sequence [31]	0.058 µM	0.64 µM	0.039 µM

## Data Availability

Not applicable.

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
