# Peer review of "Pim Kinases: Important Regulators of Cardiovascular Disease"

_ijms, 2023, doi:10.3390/ijms241411582_

Round 1

Reviewer 1 Report

The review manuscript by Nock et al. first provides basic information on the general properties of Pim family kinases, their structure and regulation, and then gets more focused by listing multiple studies, where the roles of Pim kinases in cardiovascular tissues and diseases have been described. This focus is well justified, as most Pim reviews have concentrated on the impact of Pim kinases and their inhibitors on cancer progression. While Pim inhibition might be an attractive therapeutic option for many CDV conditions, it should be considered as a double-edged sword, as PIM kinases are cardioprotective especially in cardiomyocytes, as reported in the many studies from the Sussman group, the most recent ones of which were not cited here. While mice lacking all three PIM kinases show some problems in their hematopoietic compartment, they have had time to adapt to this situation by up regulating complementary pathways, but sudden inhibition of PIM activity in CDV patients may not be as easily overcome. Thus, it should be stressed even more in the review that potential problems with cardiotoxicity should be taken seriously, especially when using pan-PIM inhibitors.

Before acceptance for publication, the English language should be carefully checked and revised and the following minor comments to be addressed:

-       In the abstract, it is sufficient to talk about Pim kinases, as this acronym (proviral integration site for Moloney murine leukemia virus) is also explained in the introduction. 

-       Repeats in the text should be avoided, e.g. in the first chapter: Signal transduction … is crucial for cell signalling; Aberrant kinase activity … may be caused by mutations, … in addition to kinase mutations.

-       Ref. 10 (Mikkers et al. 2004) is not correct in its context, as the role for PIM kinases in cell migration has been described later in Ref. 103 (Santio et al. 2010)

-       Pim inhibitors are not potential targets for chemotherapy reagents, but for cancer therapy. The term chemotherapy usually refers to the use of alkylating or other unspecifically cytotoxic agents: https://en.wikipedia.org/wiki/Chemotherapy

-       What do the authors mean with “upregulation in conditions or risk factors associated with CVD”? Should it be under conditions associated with CVD or in the presence of CVD risk factors? 

-       Figure 1 is not necessary for understanding the homology between distinct Pim family members, but if the authors wish to include it, they should explain the color coding. Also the other figures need more detailed explanations.

-       How can both overexpression and inhibition of miR-328 upregulate PIM expression?

-       How about the role of Pim autophosphorylation (especially of the longer isoform of Pim-1) for its stability?

-       Phosphorylation of the androgen receptor by Pim-1 probably not relevant for CDV

-       When talking about the single isoform of Pim3, the authors should refer to Figure 2, not Figure 1.

-       Table 2 is not very informative and should be left out, as most facts already are presented well enough in the text or in Figure 2.

-       Where do the data in Figure 4 originate from? A reference needed. 

-       How can Pim kinase deletion or inhibition offer antioxidant protection? Usually Pim inhibition causes increased ROS production.

See above

Author Response

Please see the attachment with reviewers comments in red, and responses to comments in black.

Reviewer 2 Report

The topic approached is interesting and actual. It’s a well written, well-conceived and extensively documented scientific work.

Major comments

- The general part, dedicated to structure, regulation, localization  and various biological actions of Pim kinases covers half of the manuscript – would be advisable to be restricted to the information needed to follow the part dedicated to cardiovascular effects

- Page 10 the 3rd paragraph: ‘. Quiescent endothelial cells rapidly switch to an activated state gaining the ability to sprout, migrate, and proliferate in response to VEGF released from VMSCs and macrophages and Pim-1 has been shown to mediate VEGF signalling in mice endothelial precursor cells’ – please elaborate on the effects of Pim-1 on VEGF signalling

- Table 1: Pulmonary artery smooth muscle cells MiR-206/ Pim-1/ Pulmonary artery smooth muscle cells (ref. nr. 127) – there are no results provided

- Table 2: Pim 1/ isoform differences – ambiguously presented; Sequence similarity/ Pim 2/Pim 3 – the cells are missing

- Tables with clinical trials and animal studies investigating the Pims involvement in CV pathology would be very useful

- ‘Conclusions’: the beneficial effects of Pim1 on cardiomyocytes  may worth to be mentioned

Minor comments

- The abbreviations must be defined upon first use. (e.g. page 2: ATP, Km; page 3 TNF alpha, IL etc.)

- Tables 1 and 2: the font is too big; footnotes with abbreviations requiring a definition are missing

- Figures 3 and 5: footnotes with abbreviations requiring a definition should be inserted.

- Subchapter 2.1.: announces but do not presents the localization of Pims; ‘localization’ may be excluded from this subtitle; it is presented further for each izoform

- Figure 4: a legend with concise comments and the explanation of the abbreviation (TPM) would be necessary

- Page 12, first paragraph: ‘following acute treatment of mice with a pan-Pim kinase inhibitor.’ – could you please name the inhibitor? A reference is required here as well.

- Page 12, second paragraph: ‘reports vary regarding the extent of reduced platelet count with some studies reporting no thrombocytopenia events’ please insert an appropriate reference

- Figure 5: cardiomyocyte – please insert the type of Tn, according to the information provided in the manuscript; increased SERCA activity apparently is not mentioned in the manuscript among CV influences of Pims

English language requires minor corrections. Below there are some examples:

‘The Pim kinases may therefore offer alternative therapeutic targets treat cardiovascular related diseases and comorbidities’

Page 6 first paragraph: ‘Pim-1 has also been shown to promote cell migration and chemotaxis (metastasis) the phosphorylation of CXCR4 58 .’; second paragraph ‘Similar to Pim-1, it has multiple isoforms; 34, 37, and 40kDa (Figure 2) 59 , yet it only shares 66% amino acid sequence homology…’

Page 6, second paragraph: unequal font ‘on eukaryotic initiation factor 4E binding protein (4E-BP1) and BAD, allowing for cell growth’

Page 8 last paragraph: ‘Pim-1 is increased 11-fold in coronary artery disease 68 , in pulmonary arterial hypertension 69, and Pim-2 is upregulated in atherosclerotic arteries in coronary artery disease 70 . Suggesting that Pim kinase inhibition may be a desirable therapeutic for a multi-targeted approach to prevent and treat cardiovascular diseas’

Page 11, first paragraph: ‘therefore inhibition of Pim kinases may not be a desirable therapeutic at advanced stages of plaque development, if it is leads to VSMC cell death.’

Page 14, first paragraph: ‘upregulated in in response to’

Author Response

(The authors gave the same response as above.)

Reviewer 3 Report

Dear Editors,

Thank you for the opportunity to review the paper entitled “Pim kinases: Important regulators of cardiovascular disease”. The study provides comprehensive review of the structure and functions of Pim kinases in cardiovascular diseases. 

My concern regarding some minor points includes the following.

Additional explanations should be added and the abbreviations should be expanded in captions to Figures 3 and 4.

The sentence “Pim-1 levels, suggesting ubiquitin and proteasomal regulation of Pim kinase levels” is clearly incomplete and should be edited for clarity.

The text has two sections with identical titles: 3.2.2. Endothelial cells and 3.3.1. Endothelial cells.

Author Response

(The authors gave the same response as above.)
